

# The influencing factors for distribution patterns of resident and migrant bird species richness along elevational gradients

Jin-Yong Kim[1], Jongmin Yoon[1], Yu-Seong Choi[2] and Soo Hyung Eo[3]

[1] Research Center for Endangered Species, National Institute of Ecology, Yeongyang, South Korea
[2] National Migratory Birds Center, National Institute of Biological Resources, Ongjin, South Korea
[3] Department of Forest Science, Kongju National University, Chungnam, South Korea

## ABSTRACT

The latitudinal and elevational patterns of species richness of resident and migrant birds have been of interest to researchers over the past few decades, and various hypotheses have been proposed to explain the factors that may affect these patterns. This study aimed to shed light on the elevational distribution patterns of resident and migrant bird species richness by examining biotic and abiotic factors such as climate, and habitat heterogeneity using a piecewise structural equation model (pSEM). The overall pattern of resident species richness showed a decreasing trend with increasing elevation, whereas that of migrant species richness showed an increasing trend. The mid-peak pattern of species richness was affected by a combination of resident and migrant species and not by either resident or migrant species. Our results showed that resident species were distributed in lower elevation regions with higher mean spring temperatures, whereas migrant species were found in higher elevation regions with lower mean spring temperatures and higher overstory vegetation coverage. Although high elevation conditions might adversely affect the reproduction of migrant birds, higher overstory vegetation coverage at high elevations seemed to compensate for this by providing a better nesting and roosting environment. Despite the significance of habitat diversity and understory vegetation coverage in univariate linear regression models, multiple regression models of the interconnection of ecological processes demonstrated that mean spring temperature and overstory vegetation coverage were more explanatory than other variables.

## INTRODUCTION

Latitudinal patterns of species richness and the distribution of resident and migrant bird species have been used to provide useful information on climate change over the past few decades. *Moreau (1952)* was the first to study the role of tropical Africa in the migration of Palearctic birds. In North America, *MacArthur (1959)* conducted quantitative studies on the distribution and abundance patterns of tropical migrant birds. *Herrera (1978)* found that the proportion of migrant passerine birds was increased with increasing latitude in Europe. Hypotheses regarding habitat complexity and land cover

Corresponding author
Soo Hyung Eo, eosh@kongju.ac.kr

types have been proposed, and the importance of habitat heterogeneity has been noted (*Elsen et al., 2021*). Most species have strong associations with one or more land cover types (*Elsen et al., 2021*) and specific food resources (*Willson, 1976*; *Helle & Fuller, 1988*), which can influence the bird species richness pattern. Temperature-related factors are of great importance considering that latitudinal gradients are closely associated with thermal gradients, which can influence resident and migrant species richness (*Forsman & Mönkkönen, 2003*; *Evans, Warren & Gaston, 2005*; *Elsen et al., 2021*).

Elevation-based temperature gradients are the most important factor that can influence the distribution of resident and migrant species by affecting bird fecundity and levels of parental care in breeding birds (*Badyaev & Ghalambor, 2001*; *Evans, Warren & Gaston, 2005*; *Elsen et al., 2021*). A study related to habitat heterogeneity found that the highest migrant ratios were recorded in mature upland areas and lowland areas with predominantly young or low vegetation (*Fuller & Crick, 1992*). These results are broadly in agreement with other findings showing that migratory species commonly represent a high proportion of birds in early successional habitats (*Fuller & Crick, 1992*; *Smith, Salgado & Robertson, 2001*) as the food resources make them unsuitable for resident species (*Greenberg, 1995*). These results indicated that residents having the characteristics of specialists were distributed in areas with higher habitat diversity which has a variety of habitats and food resources. Recently, there have been several studies on the latitudinal patterns of resident and migrant species richness and the influencing factors (*Smith et al., 2017*; *Elsen et al., 2021*; *Leveau, 2021*); however, limited number of studies have examined elevational patterns (*La Sorte et al., 2015*; *Katuwal et al., 2016*; *Alvarez-Alvarez et al., 2020*).

Various hypotheses related to climate, spatial factors, evolution, biology, and the elevational Rapoport's rule have been proposed to explain elevational patterns of species richness (*Stevens, 1992*; *McCain, 2009*; *Kim et al., 2018*, *2019*). Previous studies of elevational patterns of breeding bird species richness conducted in Jirisan National Park, South Korea, showed a mid-peak pattern, which tested several hypotheses (*Kim et al., 2018*) and the elevational Rapoport's rule (*Stevens, 1992*; *Kim et al., 2019*). These studies presumed that the migratory bird species would contribute to species richness. However, the heterospecific attraction hypothesis and contribution of the group with the widest range were not explained the mid-peak pattern (*Kim et al., 2018*, *2019*). And *Kim et al. (2018)* detected this mid-peak species richness pattern as a function of elevation for migratory and resident birds grouped together overall. However, it remains unclear what influences the distribution of different resident and migrant groups and how this affects the mid-peak pattern. Therefore, elevational resident and migrant species patterns is required to gain a more thorough understanding of what might explain the pattern.

In the present study, we determined the distribution patterns of resident and migrant bird species richness and examined the convergent response of different groups on the mid-peak pattern. We tested previously mentioned hypotheses regarding resident and migrant species distribution related to climate (mean spring temperature during breeding season), vertical habitat heterogeneity (understory and overstory vegetation coverage), and horizontal habitat heterogeneity (habitat diversity) using linear regression and
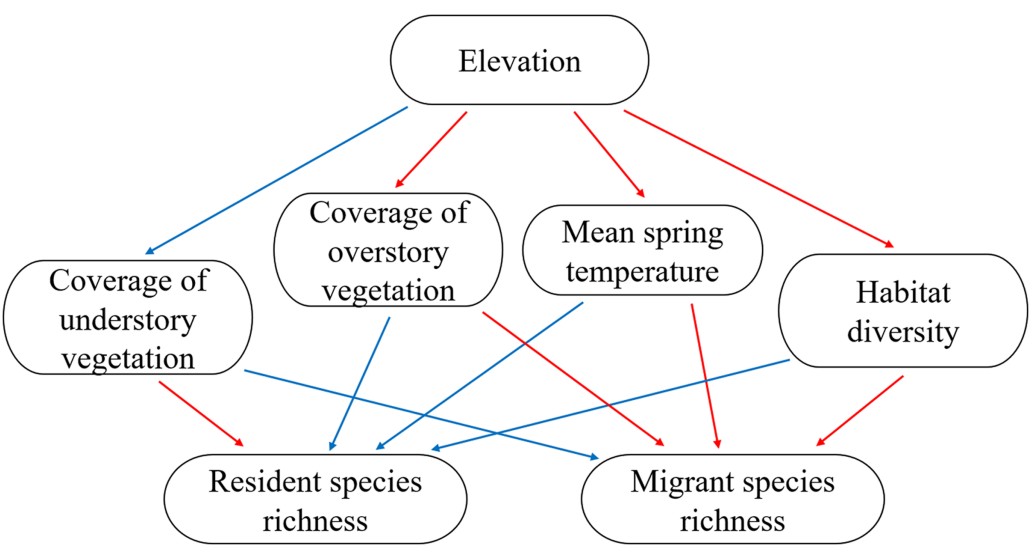

**Figure 1 The conceptual scheme of the present study.** Expectations of positive and negative relationship were indicated using blue (positive) and red (negative) arrows.

piecewise structure equation models. The conceptual scheme of the present study is shown in Fig. 1.

## MATERIALS AND METHODS

### Study area and bird survey

We used the data obtained from a bird survey by *Kim et al. (2018)*, which was undertaken in mixed or deciduous forests located within Jirisan National Park in South Korea (total area of 481.022 km$^2$) with an elevational range of 200–1,400 m above sea level (asl). The elevational range in the present study area was 110–1,915 m asl; however, we excluded subalpine forests (above 1,400 m asl), which include ridges populated by coniferous shrubs. The standardized sampling of vegetation types is important in elevational studies (*Rahbek, 1997*; *Ferreira & Perbiche-Neves, 2021*). Therefore, all field surveys were conducted only in mixed or deciduous forests. A total of 142 plots were surveyed along an elevational gradient, and we randomly chose 10–12 plots within each 100 m elevation bracket (Fig. 2). The location of each plot was recorded using a GPS device. Surveys of the bird fauna and vertical coverage of vegetation were undertaken in every plot. Point counts of birds were carried out between late May and June 2015 to account for summer migratory arrivals. Our 1-year dataset might have some uncertainties because year-to-year variations could affect species richness patterns. All breeding bird species seen and heard within a 50 m radius of each plot (0.8 ha) were recorded during the 15 min survey period. Point count surveys commenced at sunrise and ended in 1–3 h when the birds were the most active under good weather conditions (*e.g.*, without precipitation, fog, and prevalent wind). Detected birds were identified at the species level and classified as residents, migrants, and passing migrants. Passing migrant birds that were non-breeding
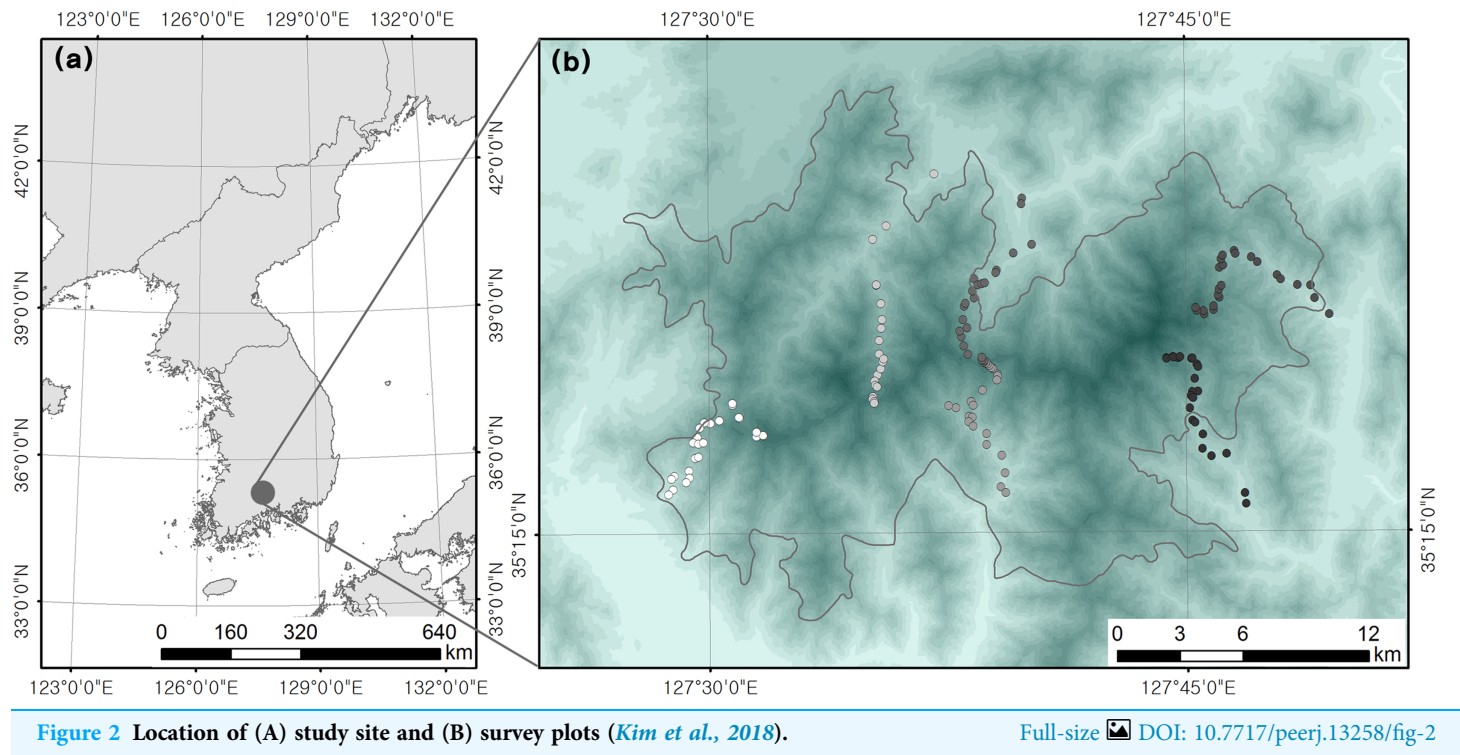

**Figure 2 Location of (A) study site and (B) survey plots (*Kim et al., 2018*).**

species were eliminated from our analyses to investigate the differences in habitat use among breeding birds (*i.e.*, residents *vs* migrants).

## Environmental variables

The Weather Research and Forecasting (WRF) model (version 3.6) was used to retrieve the mean spring temperature (*e.g.*, April to June) at regional and local scales. The maximum and minimum temperatures during the 2015 breeding season were also extracted for each survey plot using the WRF model (see also *Kim et al., 2018*, *2019*). We used the vertical coverage of vegetation as an indicator of vertical habitat heterogeneity. The vertical coverage of vegetation was surveyed at each sampling plot within a 5 m radius; vertical layers were divided into understory (less than 2 m in height) and overstory (greater than 10 m in height) vegetation with four categories in each layer: 0 (0% coverage), 1 (1–33% coverage), 2 (34–66% coverage), and 3 (67–100% coverage) (*Kim et al., 2018*, *2019*). Horizontal habitat diversity was determined by calculating the Shannon-Wiener diversity index (H′) using the area of that particular habitat type (= abundance) and the number of different habitat types (= richness), which was used as an indicator of habitat heterogeneity (*Kim et al., 2018*, *2019*). The area and number of habitat types were extracted from land cover maps (Ministry of Environment, Republic of Korea) within a 150 m radius of each plot using ArcGIS 10.3 (ESRI, Redlands, CA, USA) (*Kim et al., 2018*). A total of 15 subcategories of habitat types (residential area, commercial area, roads, public facilities, rice paddy, farmland, orchard, deciduous forest, coniferous forest, mixed forest, natural grassland, artificial grassland, swamp, barren land, water) could be found around the area and were used for the habitat diversity index.
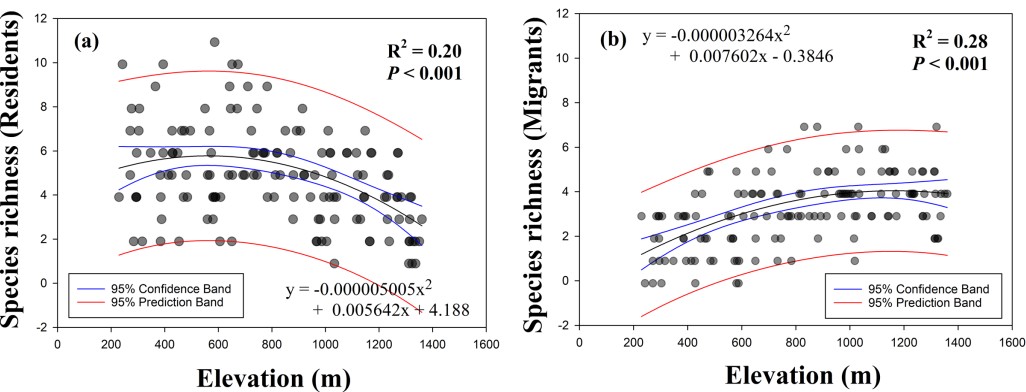

**Figure 3** Relationship between species richness (A, residents; B, migrants) and elevation.

## Statistical analyses

Two dependent variables (species richness of resident and migrant birds) and three independent variables (mean spring temperature, vegetation coverage, and habitat diversity) were used for the analysis of 142 survey plots. To determine the differences in the distribution patterns between resident and migrant birds, we analyzed best-fit curves (linear, quadratic) using $R^2$, F, and $P$ values.

We used a piecewise structural equation model (pSEM) with a generalized least squares model to test our conceptual models. The pSEM allowed us to account for the hierarchy of effects and investigate the relationship between multiple response and predictor variables (*Kim et al., 2021*). pSEMs for testing the conceptual models (Fig. 1) were constructed based on hypotheses regarding resident and migrant species distribution. Our conceptual models examined the correlations using mean spring temperature, vertical habitat heterogeneity, and horizontal habitat heterogeneity. We hypothesized the following: (1) elevation would directly affect mean spring temperature, vertical habitat heterogeneity, and horizontal habitat heterogeneity; (2) temperature, vertical habitat heterogeneity, and horizontal habitat heterogeneity would influence resident and migrant species richness. We considered spatial autocorrelation as a function of a random effect based on the coordination of each location (*Dormann, 2007*; *Kim et al., 2021*). We assessed the model (pSEM) fit to the data using Fisher's C statistics and the associated $P$ value (*i.e.*, $P > 0.05$ indicates an accepted model) (*Dormann, 2007*; *Ali et al., 2020*; *Kim et al., 2021*). All statistical analyses were performed using R 4.0.0 (packages piecewiseSEM, nlme, lme4).

## RESULTS

### Elevational patterns of resident and migrant birds

We determined the differences in elevational patterns between resident and migrant birds. The linear and quadratic patterns of single models for the species richness of resident birds as a function of elevation showed that species richness was decreased with increasing elevation (Fig. 3A). However, single models for the species richness of migrant birds as a

**Table 1 Results of linear and quadratic regressions between resident and migrant species richness along elevational gradients.**

| Responsible variable | Curve patterns | Predictor | $R^2$ | F | P |
|---|---|---|---|---|---|
| Resident | Linear | Negative | 0.145 | 23.73 | <0.001 |
| | Quadratic | | 0.197 | 17.10 | <0.001 |
| Migrant | Linear | Positive | 0.236 | 43.15 | <0.001 |
| | Quadratic | | 0.276 | 26.49 | <0.001 |

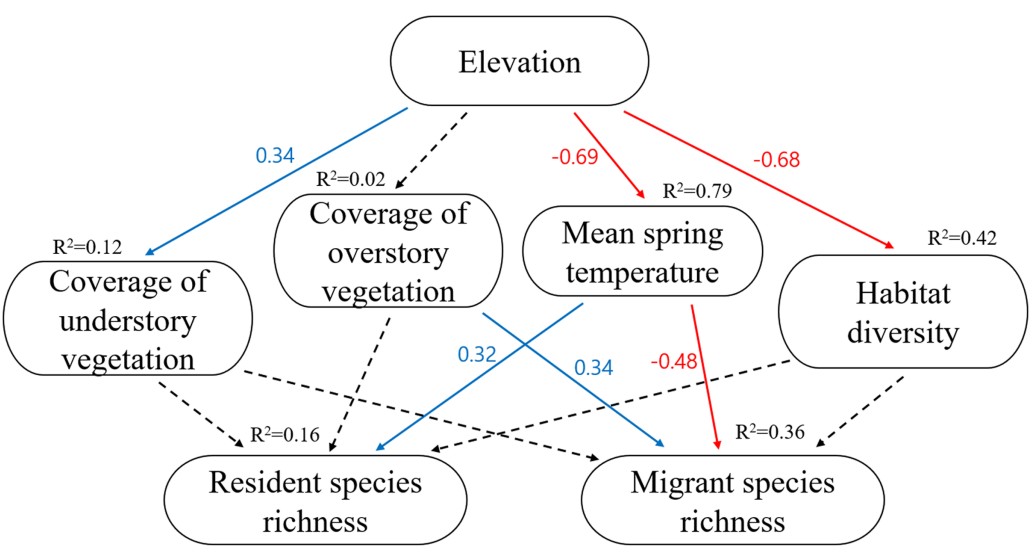

Fisher's C = 22.81; P-value = 0.198

**Figure 4 The key piecewise structural equation models (pSEM) for testing the hypotheses of resident and migrant distribution related to climate and habitat heterogeneity.** Solid blue arrows represent significant positive paths and red arrows represent significant negative paths ($P < 0.05$); dashed arrows represent nonsignificant paths ($P > 0.05$). For each variable, $R^2$ and estimate value are provided. Model-fit statistics (Fisher's C and $P$-value) for pSEM are shown.

function of elevation showed that species richness was increased with increasing elevation (Fig. 3B). For both dependent variables, quadratic patterns had slightly higher $R^2$ values compared with the values of linear patterns (Table 1) and were identified as best-fit curves.

## Factors affecting the elevational distribution of resident and migrant birds

In the pSEM (Table S1, Fig. 4), elevation had a significant positive effect on the coverage of understory vegetation ($\beta = 0.34$, $P = 0.011$) and a negative effect on the mean spring temperature ($\beta = -0.69$, $P < 0.001$) and habitat diversity ($\beta = -0.68$, $P < 0.001$). Higher mean spring temperature increased resident species richness ($\beta = 0.32$, $P = 0.025$, $R^2 = 0.16$); however, resident species richness had no significant relationship with understory vegetation coverage, overstory vegetation coverage, and habitat diversity (all

$P > 0.05$). Lower mean spring temperature and higher overstory vegetation coverage increased migrant species richness ($\beta = -0.48$, $P < 0.001$; $\beta = 0.34$, $P < 0.001$; $R^2 = 0.36$); however, migrant species richness had no significant relationship with understory vegetation coverage, habitat diversity, and resident species richness (all $P > 0.05$). The model-fit statistics (Fig. 4) indicated that the model was valid (Fisher's C = 22.81; $P = 0.198$).

## DISCUSSION

### Elevational patterns of resident and migrant birds and the convergent response of different groups

*Fuller & Crick (1992)* observed a pattern in elevational gradients, which showed that the highest migrant ratios were recorded at higher elevations. Our results were also consistent with the previously observed geographical patterns, with resident species richness having a negative quadratic relationship with elevation (Fig. 3A) and migrant species richness having a positive quadratic relationship with elevation (Fig. 3B). Previous study showed a mid-peak pattern of species richness (total species richness) using same data base (*Kim et al., 2018*). Therefore, sum of the resident and migrant species richness should be total species richness (mid-peak pattern). However, we are still unsure why elevational patterns of species richness have been found to have a mid-peak in a previous study (*Kim et al., 2018*), because our results of the richness of resident and migrant birds did not show a mid-peak. According to our results, for two intersecting quadratic curves, which meet at the mid-point, the sum of the center regions was greater than the sum of the side regions (Fig. 5). These results demonstrated that neither resident species nor migrant species singularly affected the mid-peak pattern, and the mid-regions which had the highest species richness could adequately accommodate both resident and migrant species.

### Factors affecting the elevational distribution of resident and migrant birds

Previous studies on the ambient energy hypothesis have shown that temperature is an important variable for the fecundity of breeding birds and influences the distribution of species richness (*Forsman & Mönkkönen, 2003*; *Evans, Warren & Gaston, 2005*). In the present study, the distribution of resident species in lower elevation regions was associated with higher temperatures (Figs. 3A and 4), and the distribution of migrant species in higher elevation regions was associated with lower temperatures (Figs. 3B and 4), which are in agreement with the results of previous studies (*Forsman & Mönkkönen, 2003*; *Evans, Warren & Gaston, 2005*). The ecological conditions of high elevations include colder temperatures, greater seasonality, and shorter breeding seasons that could reduce the fecundity of breeding birds and increase the amount of parental care required (*Badyaev, 1997*; *Wynne-Edwards, 1998*; *Badyaev & Ghalambor, 2001*). Therefore, high elevation conditions could adversely affect the reproduction of migrant birds.

Although migrant birds were distributed in higher elevation regions in the present study, the species richness of migrant birds showed an increasing trend with overstory

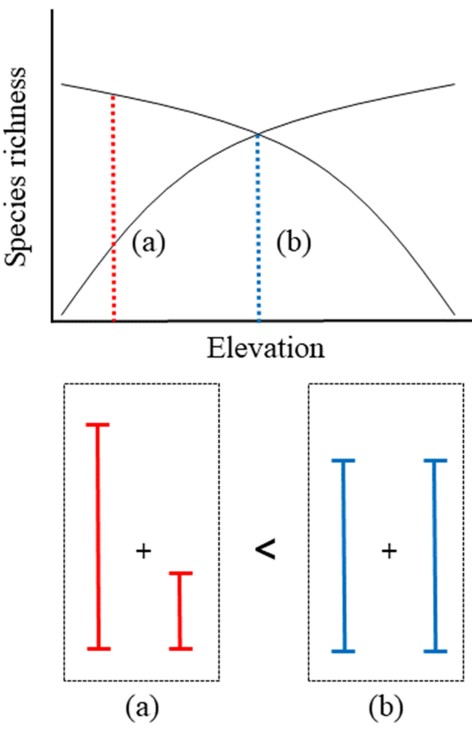

**Figure 5 Schematic representation of convergent response of two quadratic richness curves along elevation.** (A) The sum of side region, (B) the sum of center region.

vegetation coverage (Fig. 4). Regions with higher vegetation coverage could offer a mixture of resources (*e.g.*, sites for nesting and roosting and food resources) for mountain birds. In addition, habitats in these areas would likely provide considerable benefits in terms of biodiversity, especially for species threatened by climate change (*Heller & Zavaleta, 2009*; *Elsen et al., 2021*). Migrant species tolerate habitat disturbances better (*Levey, 1994*; *Smith, Salgado & Robertson, 2001*) and are more flexible than resident species in their habitat use (*Karr, 1976*; *Hutto, 1989*; *Greenberg, 1995*). Therefore, breeding migrant birds at high elevations could face disadvantageous conditions owing to low temperatures but would not experience a lack of breeding spaces and roosting sites. These results are consistent with the findings of *Fuller & Crick (1992)*, which showed that the migrant ratio was the highest in mature upland woods with little undergrowth.

Resident species are generally more specialized than migrant species; thus, the former may require narrower and more specific habitats (*Stouffer & Bierregaard, 1995*; *Smith, Salgado & Robertson, 2001*). We expected that habitat diversity would have a positive relationship with resident species richness; however, resident species richness did not show a significant relationship with the study variables except for a significant positive relationship with ambient temperature (Fig. 4). The SEM is based on the idea that systems can be controlled by networks of causal processes (*Grace et al., 2014*). In the case of univariate linear regression models that did not consider the network between variables, the results were considerably different (Table S2). In the univariate linear regression

models, habitat diversity had a significant positive effect on resident species richness and a significant negative effect on migrant species richness (Table S2). The results multiple analysis using pSEM showed that mean spring temperature and overstory vegetation had a considerable effect on the distribution of birds, and the effects of other variables were negligible.

## CONCLUSION

The elevational pattern of species richness showed a decreasing trend for resident species, whereas it exhibited an increasing trend for migrant species. The mid-peak pattern of species richness was influenced by a combination of both resident and migrant species and not by either resident or migrant species. Our results indicated that regions with the highest species richness could adequately accommodate both resident and migrant bird species. The results of pSEM analyses showed that resident species were distributed in lower elevation regions with higher temperatures. On the other hand, migrant species were distributed in higher elevation regions with lower temperatures and inhabited regions with higher vegetation coverage. Although high elevation conditions could adversely affect reproduction, migrant birds inhabiting regions with higher vegetation coverage were unlikely to experience a lack of breeding/roosting spaces and food resources for reproduction. In the univariate linear regression models, habitat diversity had a positive effect on resident species and a negative effect on migrant species. However, the results of multiple regressions, which can explain the interconnected processes of ecological systems, showed that mean spring temperature and overstory vegetation were more explanatory than other variables.

### Funding

This work was supported by a grant from Kongju National University in 2019 (2019-0266-1). The funders had no role in study design, data collection and analysis, decision to publish, or preparation of the manuscript.

### Grant Disclosures

The following grant information was disclosed by the authors:
Kongju National University in 2019: 2019-0266-1.

### Competing Interests

The authors declare that they have no competing interests.

### Author Contributions

- Jin-Yong Kim conceived and designed the experiments, performed the experiments, analyzed the data, prepared figures and/or tables, authored or reviewed drafts of the paper, and approved the final draft.
- Jongmin Yoon analyzed the data, authored or reviewed drafts of the paper, and approved the final draft.

- Yu-Seong Choi performed the experiments, prepared figures and/or tables, and approved the final draft.
- Soo Hyung Eo conceived and designed the experiments, authored or reviewed drafts of the paper, and approved the final draft.

## Data Availability

The raw data are available in the Supplemental File.

## Supplemental Information

Supplemental information for this article can be found online at http://dx.doi.org/10.7717/peerj.13258#supplemental-information.

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
