# Peer review of "The influencing factors for distribution patterns of resident and migrant bird species richness along elevational gradients"

_PeerJ, doi:10.7717/peerj.13258_

## Round 0.1 · original submission · Major Revisions

I have received two reviews with different opinions regarding your study. One of the reviewers draw attention to issues with data, analyses, writing and replicating parts of a previous study and recommends ‘rejection’. The other review points out to the need for clarification of several issues, especially regarding methods, with however, a recommendation of “minor revision”. Considering all of this, I am leaning towards ‘rejection’; but as there could be a chance for you to improve your work based on reviewer comments, as the comments by both are actually helpful in nature, I recommend a ‘major revision’.

I kindly request you to address each point raised by the reviwers and provide a rebuttal with a clear statement on if you are using previous data (please clearly mention if you have new data together with older data or not), especially with highlighting of how different the knowledge generated is from previous studies by your own group members. The language used also should be corrected to be unambiguous and succinct.

Reviewer 1 ·

Basic reporting

This manuscript describes resident and migrant bird species richness distribution patterns along altitudinal gradients in South Korea, it is generally well written and easy to read, the results are consistent with many previous studies, and I enjoyed reading it. In the manuscript, however, there are some important weaknesses that need to be addressed.

Experimental design

Methods are not clear. The authors state that they used the same methods and data as in their previous study, but this is very inconvenient for readers. Without checking the previous article, we can’t understand the experiment design. “Materials & Methods” needs to be rewriten and be clearer.
For example, lines 67-69, what is the motivation for excluding the subalpine forest data, what does “minimize the differences in bird communities among forest types” mean? If the study aims to understand the resident and migrant bird species richness distribution patterns at different elevations, why does community matter, or why this community matters that much?
Lines 63-69, how were these plots distributed at different elevations? How did the authors conduct the point counts? Were the sample efforts the same at different elevations?
Lines 79-81, how did the authors measure spring temperature (any specific time to measure or downloaded data from somewhere?), vegetation coverage, and habitat diversity? Please explain more about habitat diversity and how did the authors measure horizontal habitat heterogeneity?
Figure 4, please make it clear what are the values represent (-0.69, 0.33, 0.75, 0.26...)
In the discussion, lines 179-181, the authors first mentioned about “15 habitat types”, is it horizontal habitat heterogeneity? If it is, please provide the details in the methods.

Validity of the findings

In this study, habitat diversity has a significant effect on the resident species richness but not on migrant species richness. Rather than seeing a summary of this result (habitat as a whole), I would like to see which habitat(s) contributed more in the model, and understand well how habitats affect the two types of species richness.

Additional comments

Overall, I would like to see a more detailed and independent manuscript of this study. I hope my comments are helpful.

·

Basic reporting

The language needs to be checked (see eg the first two sentences of the abstract), the references to the literature are often approximate if not wrong, and the papers refer to mechanisms whereas it is based on correlations and only one year of data.

Experimental design

The data (already analysed in other papers) are derived from a snapshot study (one year, 15 min point counts). This will lead to high uncertainty in estimates of species richness - detectability among other sources of biases should be assessed.

Validity of the findings

The models used (structural equation models) do not bring any new information as used. The authors need to frame their questions so that the models can in fact help answering them. Right now, there is very little added value compared to previous papers published on the same data by the same authors.

Additional comments

Latitude and altitude (or rather elevation – see Körner 2007) are not variables that can affect directly species – the effects is through their associations with e.g. temperature, photoperiod, precipitation etc. This paper attempts to disentangle these effects using a survey done in in one year (2015) and already analysed in other papers by the same first author (Kim et al. 2018, 2019, 2021). As such this I a very limited dataset, particularly for migrant species that can fluctuate much between years. The difference with previous papers is the use of structural equation models (Figure 1) – but see my comment below. The paper also claims to investigate mechanisms (eg the title), but the paper only investigates correlations based on very indirect measurements (habitat diversity etc).
You write that “Herrera (1978) discovered that the number of migrant passerine birds increased with increasing latitude in Europe.” – this is wrong, Herrera investigated the proportion of migrant species, this is not the same thing (there are far fewer bird species in northern, subarctic Europe than in southern Europe). And Herrera contrasted the results obtained in Europe to those obtained in North America by MacArthur. You need to carefully assess the way you refer to the literature, it is often at best approximate.
Species richness should be estimated using one of the many methods adapted to your survey methods, to account for differences in detectability among species and habitats. As far as I could understand, there was only one point count of 15 min, so I would expect a rather large uncertainty in species richness estimates. There is a large literature on this – see Yoccoz et al. 2001 for some examples, and more recent books by e.g. M. Kéry for different modelling approaches.
You use a SEM (structural equation model), but it does not bring any new information as you could have obtained exactly the same results by fitting multiple regression models with resident/migrant species richness as a response variable and either mean spring temperature or coverage and habitat diversity as predictors, and comparing how well these models predict species richness (but this was already done in Kim et al. 2018 as far as I could judge). What could be interesting is to ask if for example temperature may have a direct effect on richness when the indirect effect on habitat variables are taken into account (investigating such direct/indirect effects is one of the main reason to use SEM or graphical models, see eg Grace et al. 2014). In other words, are the effects you estimate in models a and c completely accounted for by the effects you estimate in model b and d? Given that the models b and d do not fit, it is probably not the case. Note also that it is redundant to have elevation in all models as the coefficients by construction will be the same. Note also that you can investigate direct/indirect effects of elevation using the same tools.
You show nonlinear effects on Figure 5, but your models are all linear, right? You need to be consistent.
Check the language – for example the first sentence does not make sense with “latitudinal and altitudinal resident and migrant” – I guess you mean the latitudinal and altitudinal patterns... And the second sentence is more or less a copy and paste of the first, etc.

Nigel G. Yoccoz

Refs
Grace, J. B., Adler, P. B., Harpole, W. S., Borer, E. T., & Seabloom, E. W. (2014). Causal networks clarify productivity-richness interrelations, bivariate plots do not. Functional Ecology, 28(4), 787-798. doi:10.1111/1365-2435.12269
Kim, J.-Y., Lee, S., Shin, M.-S., Lee, C.-H., Seo, C., & Eo, S. H. (2018). Altitudinal patterns in breeding bird species richness and density in relation to climate, habitat heterogeneity, and migration influence in a temperate montane forest (South Korea). PeerJ, 6, e4857. doi:10.7717/peerj.4857
Kim, J.-Y., Seo, C., Hong, S., Lee, S., & Eo, S. H. (2019). Altitudinal range-size distribution of breeding birds and environmental factors for the determination of species richness: An empirical test of altitudinal Rapoport’s rule and non-directional rescue effect on a local scale. Plos One, 14(1), e0203511. doi:10.1371/journal.pone.0203511
Kim, J.-Y., Shin, M.-S., Seo, C., Eo, S. H., & Hong, S. (2021). Testing the causal mechanism of the peninsular effect in passerine birds from South Korea. Plos One, 16(1), e0245958. doi:10.1371/journal.pone.0245958
Körner, C. (2007). The use of ‘altitude’ in ecological research. Trends in Ecology and Evolution, 21, 569-574.
Yoccoz, N. G., Nichols, J. D., & Boulinier, T. (2001). Monitoring of biological diversity in space and time. Trends in Ecology and Evolution, 16(8), 446-453.

---

## Round 0.2 · Major Revisions

You have addressed some aspects of the improvements suggested by the two reviewers, but not to an appreciable degree. The language still remains substandard with many syntactical typos etc. Please carefully address the points raised by the two reviewers with a detailed response to each of the points.

Reviewer 1 ·

Basic reporting

I reviewed the earlier version of this manuscript and I find this revised version a considerable improvement. I appreciate the authors’ efforts to make this version more understandable. Methods of data analyses have changed and as a consequence some results have changed as well. For example, habitat diversity is not significant with pSEM, but the other main results remain the same.

However, from reading this version, I have the feeling of the authors being somewhat careless with the writing. There are many typos throughout the manuscript, and some sentences are repeated and redundant. As a minor suggestion, I suggest the authors to consistenly use either altitude or elevation in text and figures. Overall, although this manuscript has improved with the revision, it still requires a careful read and some corrections.

Experimental design

The method of vertical coverage measurement remains unclear, how did the authors measure the coverage? If the plot size is 50 m radius and all of them with two layers coverage measurement, it is a lot of work, I guess ArcGIS doesn’t provided vertical coverage.

Validity of the findings

In this version, a new hypothesis has been put forward -- interspecific competition. Although the idea has merit, I would be very conservative talking about interspecific competition in this study. First, I don’t see any direct evidence of competitions between residents and migrants, as they mentioned in the Introduction lines 30-32, the distribution pattern of residents and migrants could just be due to their habitat or food preferences. Maybe there is competition, but please provide the evidence for it. For example: how much overlap on diet, habitat preferences, etc...? This information can be found from Birds of the world: https://birdsoftheworld.org/bow/home. This could provide some useful information about potential “interspecific competition”, but this is a suggestion that the authors could consider.

Lines 214-216, I just see the distribution pattern but not “proportion of migrants and residents” in the results.

Lines 163-168, I am confused by the discussion, the authors say that altitudinal patterns of species richness have shown a mid-peak but not for residents and migrants. The following sentence says “both resident and migrant species richness repeated the fluctuation and then showed a decreasing pattern in each direction, starting at 700 m”. This is a bit self-contradictory to me. In figure 2, we can actually see a “soft” mid-peak around 700 m, but the authors call it “mid-point”, is it actually the same? By the way, in figure 2, “(a)” is missing.

Additional comments

This experiment was well designed in general as at least two papers have been published from it, and I like the attitude of authors wanting to maximize the value of the data. But I must say I was expecting to see more values from this study and some deeper thinking about the findings of the study. As it is now, the study is still limited in terms of the depth of the research questions addressed.

·

Basic reporting

The language is still poor in many places (unclear sentences, typos etc).

Experimental design

the analyses do not account for year-to-year variation or detectability.
The statistical model is poorly justified and some components are arguably wrong.

Validity of the findings

Given the limited scope of the data and the poor justification for the statistical model, the findings are unlikely to be robust.

Additional comments

I am sorry to write that this revision must have been done a bit too quickly and does not answer my main concerns.
- replacing mechanisms by causes does not make it better when the point is that you are looking at correlations based on a small dataset and you cannot assess causes based on this. You may say that your statistical model is helpful in making predictions in a given context (and since you have only one year of data the validity of such predictions is hard to assess). Going from a SEM to assessing causes require both more data and evidence that the structure of the model may reflect the actual causal mechanisms. For example altitude (you should use elevation – see Körner’s various papers) has indeed a causal relationship with temperature (the physical basis is well-known), but the causal relationship with vegetation (different from the effect of temperature as your figure seems to imply) is far from being clear – and seems even wrong (ie spring temperature is related to vegetation).
- You have now more or less copied and paste the description of the survey methods from Kim et al. (2018). But regarding the uncertainty in the description of bird communities, you only write “Therefore, our one-year dataset might possess an uncertainty that year-to-year variations were not verified in the present study.”. Besides the fact that the sentence does not really make sense, the problem is that this year-to-year variation exists (not that it might exist) and that it is known to be large in most bird communities. Moreover, standardization of methods helps in reducing variation, but it does not suppress it or remove biases due for example to different species detectabilities. I have participated myself to field campaigns monitoring bird communities and the variation from day to day can be large, even if the counts are made at the same time, with the same weather etc. Just design your study so that you can account for it (e.g. by having repeated counts). There is now a very large literature on bird communities based on occupancy models showing that even with highly standardized data, detectability is an important issue.

There are still many typos that reflect that the revision was done too quickly – eg in the abstract:
l. 27: higher overstoy vegetation
l. 28: birds, , higher overstory
In the changes made l. 82-83 of the word document:
“are poplated by coniferous shrubs. Because, standardized sampling effort per vegetation types are an importnat part of altitudinal studies”
etc.

---

## Round 0.3 · Major Revisions

I have yet again received the recommendations from two reviewers, including one new reviewer. They both recommend major revision. Please carefully address the issues that they raise and provide a clear rebuttal of issues that you don't agree with, so that we can make the final decision on your paper soon.

·

Basic reporting

Some sentences are still unclear or have a meaning different from what is intended, for exemple:
End of abstract “demonstrated that mean spring temperature and overstory vegetation coverage were more explanatory than other variables”. What do you mean? A variable is explanatory or not, do you mean that they explain more of the variation in species richness? As all non-native English speaker, we struggle with the language, but it should not prevent aiming at clear sentences.
l.61: “however, limited studies have discussed examined altitudinal patterns…”. I guess you do not mean that the studies were limited (that is lacking rigor or coverage), but that there were a limited number of studies.
You have removed causes from the title and abstract, but you still refer to causal relationships in the text, e.g. l 133 ff (or Table S1 and S2):
«Our conceptual model models examined the causal relationships among mean spring temperature, vertical habitat heterogeneity, or and horizontal habitat heterogeneity: 1) altitude would directly affect mean spring temperature, vertical habitat heterogeneity, and horizontal habitat heterogeneity”
Altitude may have a causal effect on temperature but is unlikely as such to have a causal effect on heterogeneity (i.e. as you explicitly mention temperature, do you mean it would be through another mechanism, ie atmospheric pressure, precipitation, forest use?? See Körner 2007). You are exploring correlations (that could be partial etc – as in the original papers by Wright), but don’t call them causal when there is no causal mechanism underlying them and well supported by available data.
Note also that the differences between significant and non-significant regression parameters (ie your tables S1 vs S2) are not due to using a SEM but comparing multiple regressions to univariate linear regressions.
Check the language for example L 163: “Migrant species richness was decreased with mean spring temperature and increased with overstory vegetation coverage”. Something like “Higher mean spring temperature and overstory vegetation coverage increased migrant species richness”. Check the whole results paragraph.
In Tables S1/S2, no explanation is given of what is in the tables (beta, critical value, standard estimate)
The paper by Körner about using elevation rather than altitude is McVicar and Körner (2013). Many people still use altitude and not elevation but I agree that elevation is probably the right word to use.
Körner, C. 2007. The use of ‘altitude’ in ecological research. Trends in Ecology & Evolution 22:569-574.
McVicar, T. R., and C. Körner. 2013. On the use of elevation, altitude, and height in the ecological and climatological literature. Oecologia 171:335-337.

Experimental design

See previous comments - the authors now acknowledge that for example one year is a limited study, but with a rather strange sentence "Therefore, our one-year dataset might possess some uncertainties as year-to-year variations were not verified in the present study". You cannot "verify" year to year variations.

Validity of the findings

There is confusion regarding the differences between the simple linear regressions and the multiple regressions underlying the results of pSEM. Also the causal language needs to be removed .

·

Basic reporting

Although I found this work interesting, I feel there are still a few areas that need clarity whether because of English language improvement or because just a clearer explanation is needed.

Parts of the introduction could be improved by specifying the nature of the relationships (that is the results) of the multitude of studies mentioned so that it’s not just a list. For example:
L52 how do temperature gradients influence distribution of resident and migrant spp?… the next sentence immediately transitions into a different factor completely so that nothing is learned about the relationship/actual result.

L58 “…as the food resources make them unsuitable for resident species” Why? As is, this sentence is confusing.

L69… Confusing sentence, suggested improvement “However, the hypotheses tested, which include the attraction hypothesis that states ???? and wide-ranging bird groups, were not applicable to bird species distribution along the altitudinal gradient.”

L79 modify what is in parenthesis to: “understory and overstory vegetation coverage*. Otherwise, it is confusing why coverage would represent vertical heterogeneity when it is inherently a 2-dimensional variable.

L80 add “…(habitat diversity) using linear regression and piecewise structural equation models.” To specify the methods used.

L80. You state that your conceptual model is shown in Fig1. THIS IS INCORRECT. As is, the conceptual figure is currently Figure 2. I suggest you move your conceptual figure to indeed be figure 1, so it corresponds with the text. I was a bit surprised to encounter this type of careless error at this stage in the review process.

Experimental design

I am surprised that the authors did not include a map of the study area showing the plots (experimental design) so that readers better understand this aspect (and also appreciate the amount of field work that seems to have gone into this study). This is especially because you already have GPS coordinates.

L85 replace ‘performed’ with ‘undertaken’ and add ‘located within Jirisan National Park…”
L88 end of that sentence should read “which include ridges populated by coniferous shrubs”
L115 delete “(richness)” because richness always refers to number of species not to different habitat types.
L119 It is very strange to include roads as a habitat type. Perhaps add land use/habitat types to refer to the subcategories.
L135 Make the : into a period to end the sentence. I suggest starting the next sentence with “We hypothesized the following: 1) altitude….”

Validity of the findings

L176 “we are still unsure why altitudinal patterns…” This is not sufficient. Can you offer explanations based on your analysis of that study and the literature.
L178 did you intend to finish the sentence clarifying that the absence of the mid-peak in this study? If so, modify to “…did not show a mid-peak in this study.”
L180 This sentence could be clarified by modifying as follows: These results demonstrated that neither resident species nor migrant species singularly affected the mid-peak pattern, and the mid-region which had the highest species richness could adequately accommodated both resident and migrant species.
L218 – 221 You state the main result of the linear regression models independently from the pSEM. Add a sentence that consolidates these results, given you state in L217 that the results of the two modeling approaches “were considerably different”.
L221 that states “the pSEM analyses showed that temperature had a considerable effect on dist of resident species, and the effects of other variables were negligible.” Seems to be incomplete compared to your last sentence of the conclusion L236 that states “However, the results of pSEM analyses, …showed that mean spring temperature and overstory vegetation were more explanatory than other variables.” So is overstory vegetation as an influential factor missing in L221?

Additional comments

In your abstract, your first words read a bit odd, I recommend you replace the word ‘or’ with ‘and’ to read “Latitudinal and altitudinal patterns….”

Table 1 Include an additional column with the functions/equations of the models you used or at least the important coefficients. Alternatively, include the equation in your figure showing the ‘relationship between BIRD species richness and altitude.

Fig of the Conceptual Scheme – To avoid repetition with Fig 4 of the pSEM, I only see value in retaining this figure if you represent your hypotheses, ie. your expectations of positive or negative relationships, by adding a + or – sign on the arrows or showing arrows for positive expected relations in black and negative expected relationships in red (or grey).

Figure 3 Altitudinal patterns of resident and migrant spp richness. Overall, a nice figure, but can you explain the meaning of the dots in your figure text.

Figure 4, modify the last sentence of your figure text description to “Model-fit statistics (Fisher’s C and P-value) for the pSEM are given.” A note about the arrows, typically the magnitude of the standardized regression coefficient is represented by the thickness of the arrows. Can you adjust your arrow thickness to reflect this? If you will be using color in your figures, might as well make the negative relationships red or grey and the positive relationships black as is typically done for pSEMs.

Figure 5 Consider modifying figure text description to something like “Schematic representation of convergent response of two quadratic richness curves along altitude to help visualize the altitude with peak richness.” I also recommend distinguishing which curve is for resident bird species and which is for migrant bird species. This can be done by directly labeling, changing the line style or using color to distinguish the curves.

---

## Round 0.4 · Minor Revisions

You have addressed most of the reviewer concerns thus far, but one of the reviewers have a few minor revisions suggested. Please address these, and check carefully for any syntactical errors in possibly the final revision before acceptance.

·

Basic reporting

The authors have mostly answered my previous comments, and the language has been improved.

Experimental design

The authors refer to the limitations of the study, and the methods are sufficiently well described

Validity of the findings

The data are limited but can still contribute to a wider assessment of associations between elevation and bird communities

·

Basic reporting

The manuscript is improved. But, I feel there are still a few areas that need clarity whether because of English language improvement or because just a clearer explanation is needed. It was unfortunate that the line numbers in the rebuttal did not match the lines in the new manuscript pdf. In the future, just accompany your explanation/justification with a copy of the modified sentence into the rebuttal letter directly.

L52 It is good to see implementation of the recommendation. For clarity, please modify your sentence to read: “Elevation-based temperature gradients are the most important factor that can influence the distribution of resident and migrant species by affecting bird fecundity and levels of parental care in breeding birds (Evans, Warren & Gaston, 2005; Elsen et al., 2021; Badyaev & Ghalambor, 2001)”

L71 elevatinal – correct your spelling

L73-76 suggested for clarity: ”However, this heterospecific attraction hypothesis and contribution of the group with the widest range were not applicable to bird species distribution along the elevational gradient, nor towards explaining the mid-peak pattern.”

L76 to 79 Having gone back to the original Kim et al 2018 paper, I now realize that difference between the previous study and this one. I think that you need to explicitly mention somewhere in the last two paragraphs of the introduction that Kim et al detected this mid-peak spp richness pattern as a function of elevation for migratory and resident birds grouped together overall. In this paper, it seems like you all are now dividing that data into two groups: the resident spp and the migratory spp to “gain a more thorough understanding of what might explain the pattern”.

Experimental design

I still feel a simplified version of your original map, or at least tell the reader “for a map refer to Kim et al 2018. Having in this m.s. directly could help orient the readers to the experimental design of this work (and just generally to help orient readers without having to go digging).

Validity of the findings

L182 fix word order from ‘quadratic negative relationship’ to ‘negative quadratic relationship. Do the same for the positive one.

L186 I think this first paragraph of the discussion is really confusing. Since two of the authors are in both papers and the same dataset was used in both papers, it seems that you all should be best poised to resolve “why elevational patterns of spp richness were found to have a mid-peak in the previous study…” but not in the current one. I still think it is not sufficient to say ‘we are unsure why the difference…’.

L189 You state “According to our results, for two intersecting quadratic curves, which meet at the mid-point, the sum of the center regions was greater than the sum of the side regions” Doesn’t this result explain and support the mid-peak pattern identified in Kim et al 2018, where the grouped data resulted in the hump? It seems to me like this indicates some support of that mid-peak pattern (and your figure 5 explains the logic of why this is so). So I remain confused to why you keep saying that the results of the previous study and this one are contradictory.

Additional comments

Figure 3 You have still not stated what the meaning of the dots in your figure text. Do these signify significance? If so, at what p value, etc.

---

## Round 0.5 · accepted · Accept

You have addressed the reviewer comments to an appreciable degree and the manuscript now meets the editorial requirements for acceptance. I wish you the best with your future research.